# High oxygen barrier packaging materials from protein-rich single-celled organisms
Kiran Reddy Baddigam [1] ✉, Bor Shin Chee[2], Elodie Guilloud[1], Chaitra Venkatesh[2], Helena Koninckx[3], Kim Windey[4], Margaret Brennan Fournet[2], Mikael Hedenqvist [1] ✉ & Anna J. Svagan [1] ✉

Fossil-based packaging materials pose significant environmental challenges due to their persistence and carbon footprint, resulting in pollution and long-term climate change. Here we develop bioplastic packaging alternatives (films and trays) from protein-rich microbial biomass with glycerol as the plasticizer. The microbial biomass demonstrated excellent film-forming properties through compression molding, and the final materials exhibited good mechanical properties and excellent gas barrier properties - an average oxygen permeability coefficient of 0.33 $cm^3$ mm $m^{-2}$ $day^{-1}$ $atm^{-1}$ at 50% relative humidity and 23 °C. The oxygen barrier properties highlight these microbial biomass materials as a promising, sustainable alternative to fossil-based synthetic films like EVOH, which are widely used in multilayer food packaging. Beyond offering a microplastic-free solution, the protein-rich materials present an opportunity to mitigate microplastic pollution at the end of their lifecycle. The current results position bioplastics based on microbial biomass as a critical step forward in addressing environmental sustainability challenges with current commercial packaging materials.

Plastics have rapidly proliferated across many application fields due to their favorable qualities, such as out-matched mechanical, barrier and chemical properties, as well as processability, and low production costs[1–4]. Since large-scale plastic production began in the ≈1950s and up to 2015, an estimated 8300 million tonnes of virgin plastic have been produced. Regrettably, more than half of this quantity (5000 million tonnes) has ended up as waste in oceans and other natural environments[5]. It is only a few types of plastics[3]—polyethylene (PE), polypropylene (PP), polyvinylchloride (PVC), poly-ethylene terephthalate (PET), polyurethane (PUR), polystyrene (PS), aliphatic polyesters and polyamides—that account for most plastics (92%) ever made between the 1950s and 2015[5]. The primary drawback of our common plastics is that the monomers are mostly derived from fossil hydrocarbons which are linked to problems that include climate change and biodiversity loss, furthermore, most fossil-based plastics are not biodegradable and hence can accumulate in the environment[6,7]. The largest market for plastics is as packaging materials[8,9], but packaging materials unfortunately have a relatively short product life span[5]. After their use, much of the plastics are still, in Europe alone, disposed of without entering a sustainable and circular life cycle[9]. In Europe, packaging accounts for 39% of plastic usage[9], and with single-use plastics comprising 50% of the plastic fraction (80−85%) of marine litter[10]. Due to the plastics' slow decomposition rates, they undergo long-term fragmentation when discarded in natural environments, with polluting secondary microplastics being released

through photo-, physical, and biological processes[7]. According to the UN Environment program, an estimated 11 million tonnes of plastic generated on land enter the world's oceans every year[11].

Ethylene vinyl alcohol (EVOH) is a moisture-sensitive thermoplastic known for its outstanding oxygen barrier properties at low relative humidities due to the presence of -OH in its molecular structure[12]. In food packaging, EVOH extends the shelf-life of oxygen-sensitive foods by limiting the presence of oxygen and hence slowing down the degradation. For the oxygen barrier to be effective, EVOH must be sandwiched in a multilayer structure with other plastics, e.g. PE or PP, to protect it from moisture[12,13]. The disadvantages of EVOH include high carbon footprint, high production costs, and longevity, making it less economical. In particular, the multilaminar structures reduce the recycling possibilities of EVOH and, as a result, at the end of its life, EVOH is often incinerated or sent to landfills, where its biodegradability remains minimal[14].

Protein-based biopolymers are bio-based and biodegradable alternatives to EVOH. Pure protein materials are inherently stiff, often brittle and have quite high barrier properties towards gases in low to moderate moist conditions. This is due to the low molecular mobility caused by the stiff peptide bond, extensive dipolar interactions, hydrogen-bond network and the presence of bulky amino acid side groups[15]. This makes proteins, from a variety of plant and animal

[1]KTH Royal Institute of Technology, Dept. of Fibre and Polymer Technology, Stockholm, SE-100 44, Sweden. [2]TUS Technological University of the Shannon – Midlands Midwest, Centre for Polymer Sustainability, PRISM Research Institute, Athlone, Ireland. [3]Avecom nv, Wondelgem, Ghent, 9032, Belgium. [4]Valpromic nv, Wondelgem, Ghent, 9032, Belgium. ✉e-mail: Kiranreddy.baddigam@gmail.com; mikaelhe@kth.se; svagan@kth.se

**Table 1 | The molecular composition of protein-rich bacterial biomass**

| Parameter | Microbial Biomass |
|---|---|
| Dry solids (%) | 92.5% |
| Total Fat (% - DS) | 5.4% |
| Protein (% -DS) | 68.4% |
| Carbohydrates (% -DS) | 12.2% |
| PHBV (% -DS) and (HB:HV composition) | 2.8% and (0.6:0.4) |
| Ash (%-DS) | 7.3% |
| Calcium (%-DS) | 0.2% |
| Potassium (%-DS) | 2.3% |
| Sodium (%-DS) | 0.1% |
| Total Phosphorus (%-DS) | 1.3% |

DS – dry solids, values are based on a DS basis.

sources, suitable as gas barrier layers in i.e. packaging. Examples of previously studied protein-based materials include wheat gluten, soy protein, bean, whey, collagen and keratin proteins[15]. A newer, but still underexplored, protein source is protein-rich microbial (yeast, bacteria, microalgae, fungi) biomass[16–20]. The advantages of such biomass include rapid growth rates, minimal land- and water requirements, a low carbon footprint, and year-round flexibility, ensuring a steady supply of biodegradable material[16,17]. The sustainability of microbial biomass is influenced by the feedstock (organic wastewaters, agro-industrial side-streams, $CO_2$ etc)[16,17,21] and both aerobic and anaerobic growth conditions have been investigated[21], further enhancing its potential as a versatile and eco-friendly resource[22]. Bjurström et al. developed microbial biomass-based films and evaluated their mechanical properties, moisture uptake behavior, and biodegradability in the context of circular bioeconomy applications. However, the films exhibited poor mechanical performance due to low internal cohesion, and the authors highlighted the importance of employing the correct pre-processing step during preparation[17].

In this study, we explore the potential of developing mechanically improved packaging materials derived from protein-rich (68%) bacterial biomass produced using liquid side streams from the potato processing industry. To evaluate the processability of this biomass, a prototype tray and thin films were fabricated. These were intended to function as potential oxygen barrier layers in multilayer packaging or as single layers in dry food applications. The biomass-based materials were thoroughly characterized for their oxygen barrier and mechanical properties, and their performance was then benchmarked against EVOH, highlighting their feasibility as sustainable alternatives in packaging applications.

## Results and discussion
### Protein-rich Microbial biomass (MB)
Herein microbial biomass, in the form of spray-dried protein-rich multi-strains of bacteria, was exploited in the preparation of the packaging material prototypes. The bacteria were cultivated using liquid side streams from potato processing. As expected the composition of the biomass, summarized in in Table 1, contained a variety of biomolecules including proteins (68.4%), lipids (5.4%), carbohydrates (12.2%), and a small amount of PHBV (2.8%). The protein fraction was quite high, which is typical for a protein-rich bacterial biomass[23]. The lipid composition and amino acid composition are provided in Supplementary Table 1 in the SI file. We note that the amino acid composition is similar but not identical to that of a previous high-protein material based on a bacterial mixed microbiome[16]. The composition of amino acid building blocks influences the chemical structure and material properties, which include mechanical properties, water uptake, barrier properties and biodegradability[24–26].

### Prototype films and trays from protein-rich bacterial biomass
The spray-dried biomass was processed into films and trays by compression molding, the different processing steps are summarized in Fig. 1a. The morphology of the starting spray-dried biomass and the final materials are presented in Fig. 1c (Supplementary Movie 1 and Supplementary Movie 2). The SEM images of MB powder (Fig. 1c, I) revealed a granular powder structure. Notably, no visible microbial organism shapes could be identified, neither in the MB powder nor in the biofilms and the tray, suggesting that the spray-drying step effectively disrupted the bacteria. The average particle size of powder particles was ~20 μm. All types of processing, here spray-drying and hot pressing, are important for the final physicochemical properties of protein-based materials. This is because they disrupt molecular interactions and contribute to the unfolding of protein structures and the rearrangement of protein chains into new three-dimensional networks[27]. The present study is, to our knowledge, the first to use a commercially viable spray-drying step in the preparation of microbial biomass materials. Techniques including oven drying[16,17] have previously been utilized, which generated MB flakes instead of particles. Additionally, MB in water has directly been homogenized and solvent-casted to generate films[18,28,29]. Here a plasticizer, glycerol, was added in the preparation of films and trays, which increased chain mobility and reduced the brittleness of the present high-protein materials. Although various plasticizers can be used with proteins[30], we found, based on a large set of potential and known plasticizers, that glycerol was the most effective one for wheat gluten protein. Hence, it was used in the present study[31]. Unless specified otherwise, the films and tray were formulated with 25 wt% and 20 wt% glycerol, respectively. Initially, the tray was fabricated using a 25 wt% glycerol-blended MB material. However, that tray lost its original shape within a month, indicating the need for increased stiffness. By reducing the glycerol content to 20 wt%, a tray with sufficient stiffness over time, was successfully produced.

The optical properties of the films are presented in Fig. 1d, with the optical properties of a PET film included as a comparison. A film that has an optical transmission >90% at 600 nm is transparent to the eye[32]. The present MB film had a total transmittance of 23% at 600 nm, making it significantly less transparent compared to that of a PET film. However, the MB film was also able to block UV light, with transmittance values of ~ 0% between 200−400 nm. This property is attractive in certain packaging materials that are designed to protect photosensitive food products, such as dairy products, from harmful UV radiation[33].

Thermal gravimetric analysis (TGA) measurements of neat MB powder, a glycerol-plasticized MB-based tray and film, further supported the presence of glycerol, observed as greater weight-loss with increasing glycerol content at temperatures above ~150 °C. For all three samples, the most significant weight loss was observed before ~ 270 °C, which is consistent with the thermal properties of other types of glycerol plasticized protein-rich microbial biomass materials[16].

We observed that our plasticized MB films were tacky, which is interesting as it can be used to create easily sealable/resealable packaging. As a demonstrator, we prepared a pouch containing low-density polyethylene pellets by folding and gently pressing the short edges of the formed pouch, as shown in Fig. 2a. As the seams are not welded in Fig. 2a, the joints were not permanently sealed and the pouch could be re-opened by peeling the two film pieces apart. Additionally, as a demonstrator, two pieces of MB films were sealed together using a standard heat-sealing equipment, images after sealing are included in Fig. 2b.

### Chemical structure of the microbial biomass materials
The physical and chemical modification of the initial protein-rich microbial biomass will influence the properties of the final materials. Here FTIR was used to understand the chemical changes as the MB powder was pressed into films and trays, and in particular the important secondary structure of the protein in the final materials because proteins were the largest constituent (Table 1) in the microbial biomass. These proteins, in accordance with other Intrinsically Disordered Proteins (IDP), contain a significant

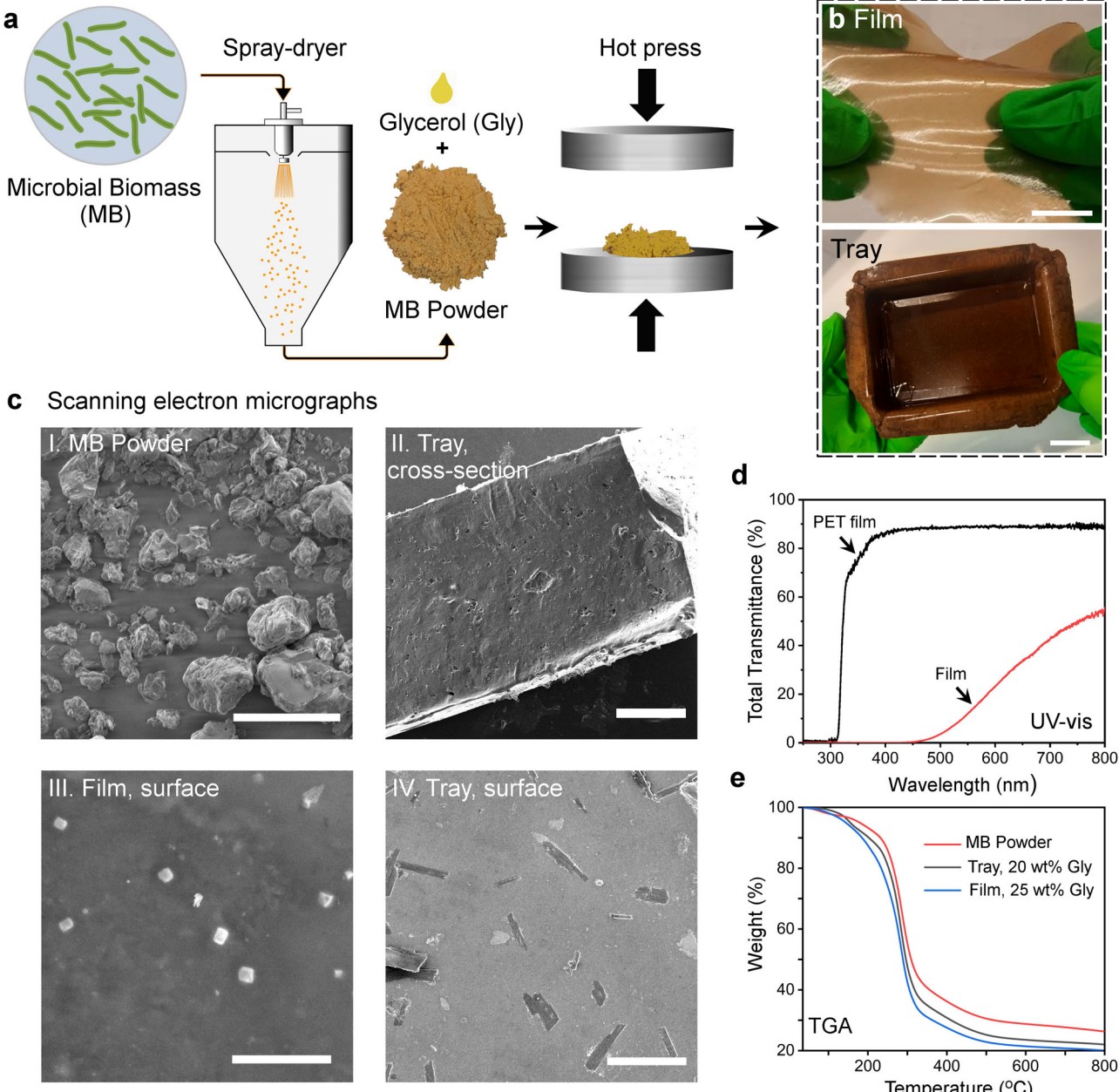

**Fig. 1 | The preparation steps, morphology and optical properties of microbial biomass based materials. a** Spray-dried microbial biomass (MB) is converted into (**b**) films and trays by hot pressing. Plasticizer content: 25 wt% (Film) and 20 wt% glycerol (Tray). Scale bars: 2 cm. **c** Scanning electron micrographs of (I) the spray-dried MB powder, (II) cross-section of the MB tray, and the surface of a (III) MB film and (IV) MB tray. Scale bars: 100 μm (I), 500 μm (II), 5 μm (III) and 25 μm (IV). **d** The total transmittance of a MB film (thickness 150 μm, red line), compared with a PET film (100 μm, black line). **e** Thermal gravimetrical analysis of MB powder, (red line) a MB tray (black line) and MB film (blue line), with increasing amounts of glycerol (Gly).

amount of unordered feature-less structures as well as self-assembled ordered secondary structures, such as α-helices, β-sheets and β-turns[34]. The FTIR spectra of the starting spray-dried MB powder with 25 wt% added glycerol, MB film (25 wt% glycerol) and tray (20 wt% glycerol) are presented in Fig. 3. In Supplementary Fig. 1 in the Supplementary Information, FTIR spectra are included for the MB films with different glycerol contents (20, 25 and 30 wt%). A difference between the MB powder and the hot-pressed film was primarily found in the amide I band region, 1700–1580 cm$^{-1}$. The deconvoluted and resolved FTIR spectra are shown in Fig. 3b–d, and Supplementary Fig. 1b–d. By hot-pressing the MB powder/glycerol at 110 °C for 5 min an increase of β-sheets (1618 cm$^{-1}$ and 1625 cm$^{-1}$), with strongly hydrogen-bonded peptides, were observed, compare the relative sizes of the different components in Table 2. This observation aligns with a

previous study[16]. Increasing the glycerol content (20–30 wt%), however, did not significantly enhance the amount of strongly hydrogen-bonded β-sheets (Supplementary Fig. 1 and Supplementary Table 2 in SI). An increased content of β-sheets can lead to increased stiffness and strength, particularly if arranged into nanofibrils[35]. The changes in secondary structure during a heat operation are also a consequence of the chemical changes occurring on an amino acid level. Disulfide-thiol interchange reactions, leading to polymerization and further crosslinking, is a commonly occurring feature at elevated temperature. Especially the content of both lysine and tyrosine was quite high in the MB (Supplementary Table 1), hence, isopeptide formation (involving lysine) and dityrosine crosslinking likely occurred during the hot pressing operation[36]. It should be noted that the FTIR spectra of the MB (see Fig. 3b–d) contained an additional strong band centered at ca. 1595 cm$^{-1}$,

**Fig. 2 | The microbial biomass film as a potential sealable/resealable packaging material. a** A glycerol-plasticized (25 wt%) microbial biomass (MB)-based film (7.4 cm × 7.4 cm) is tacky and can be folded into a pouch by gently pressing edges together. The pouch can be re-opened, because the edges are not permanently sealed. **b** Two MB film pieces were sealed together by applying pressure and heat.

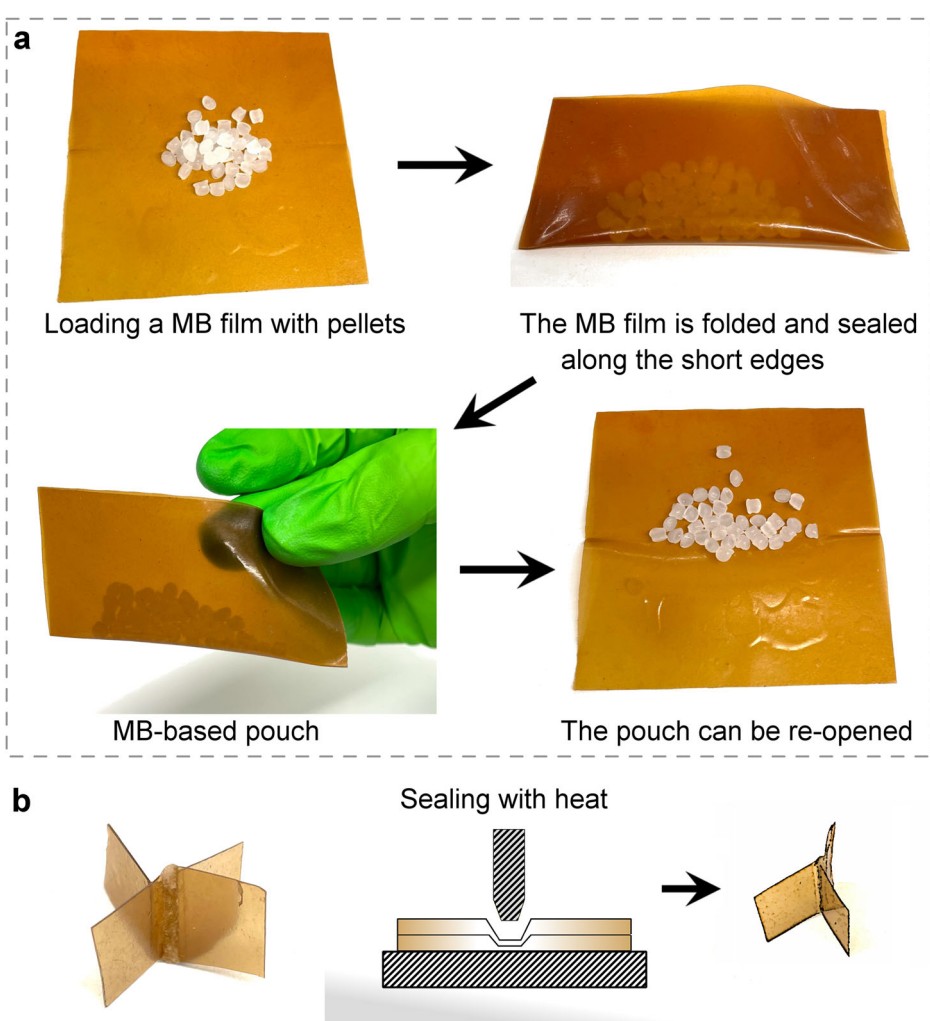

which is normally not observed in the Amide I region for proteins. Therefore, it was not included in the secondary structure analysis.

When the pressing time was extended from 5−25 min for tray formation, a decrease in strongly hydrogen-bonded β-sheets (with an increased amount of weakly hydrogen-bonded β-sheets) was observed (Table 2), as indicated by a comparison of relative peak areas at 1618 cm$^{-1}$ and 1625 cm$^{-1}$ between the MB film (20 wt% glycerol) and the MB tray. This suggests that longer heat treatments may potentially negatively impact the mechanical properties of microbial biomass materials.

### The mechanical properties of microbial biomass materials

The mechanical performance of the produced biopolymer film and tray is summarized in Fig. 4. The tensile stress-strain curves of the materials exhibited a typical elastoplastic behavior; the film with 25 wt% glycerol had an elongation at break of 67 ± 8% (mean ± SD), while the tray, containing 20 wt% glycerol, demonstrated a slightly higher strain at break of 87 ± 23% (Fig. 4a, b). There was no significant difference between the Young's moduli, whereas the maximum tensile stress values were slightly higher for the tray. This is despite the extended heat treatment time for the tray. The mechanical properties depend on the processing conditions, as exemplified previously[16]. In this study, the pressing temperature for the film and tray was set to 110 °C, with pressing times of 5 min for the film and 25 min for the tray. Singha et al.[16] reported a decrease in stiffness for protein-rich single-cell materials when the compression time was increased from 5−15 min, particularly at the highest tested pressing temperature of 130 °C (110–130 °C tested). An opposite trend was observed for the maximal tensile stress, which increased with longer pressing times at 110 °C. These trends align well with the results

presented in Fig. 4b, c. The elongation at break for the present materials was quite high, an order of magnitude higher than previous materials based on single-cell proteins[16,17] or yeast cells[29]. The processing conditions influence ductility[16,17], and we hypothesize that the pre-processing (spray-drying step) further influenced the cohesive properties of the MB films. Indeed, the spray-drying step appears to have efficiently ruptured the microbial cell membranes, eliminating most traces of intact microbes (Fig. 1c).

The reported Young's modulus (E) and maximum tensile strength values surpassed those of previous studies on films made from yeast cells[29], or single-cell proteins[16,17]. The elastic moduli (65 ± 8 MPa and 55 ± 4 MPa, mean ± SD), maximum tensile stress (1.9 ± 0.1 MPa and 2.1 ± 0.1 MPa, mean ± SD), and elongation at break for the MB-based film and tray are comparable to those of plasticized protein isolate films, e.g. whey protein isolate films (E ≈ 50 MPa, maximum tensile strength ≈4 MPa and elongation at break ≈100%)[30]. Yet, the stiffness and tensile strength were approximately an order of magnitude lower compared to those of fossil-based plastics (e.g. PE, PP). However, when compared to natural materials, these mechanical properties were better than certain types of soft tissue, and close to the mechanical properties of low-density cancellous bone, see Ashby-plots in Fig. 4e. This suggests that for applications requiring enhanced mechanical strength, MB films should be incorporated into a multilayer structure, analogous to the configuration typically employed with EVOH.

### Oxygen Barrier properties of microbial biomass materials

Protein films are highly valued for their excellent oxygen barrier properties, making them ideal for certain food packaging and preservation

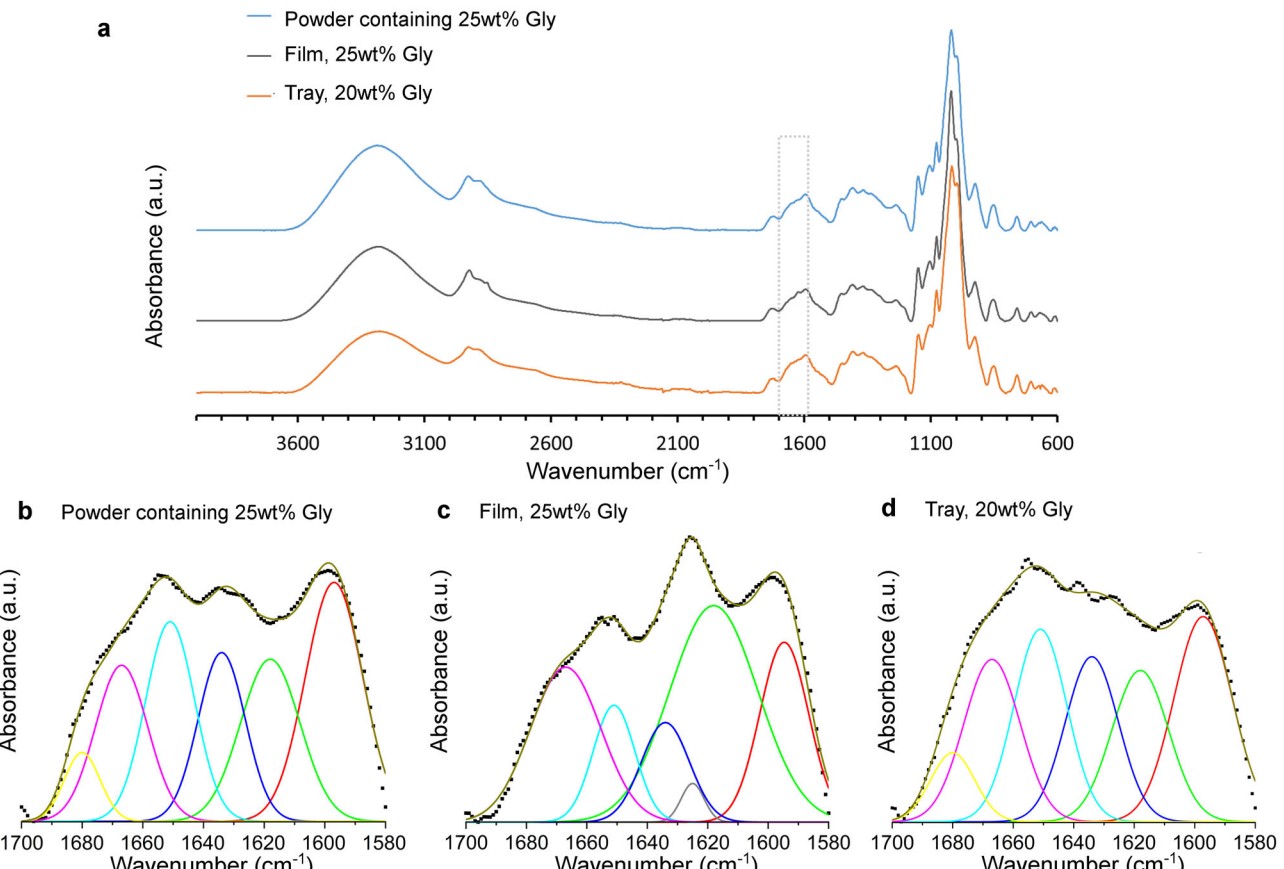

**Fig. 3 | The chemical structure of microbial biomass (MB) materials.** FTIR spectra for (**a**) the MB powder with 25 wt% glycerol (Gly, blue line), a MB film (25 wt% Gly, black line) and MB tray (20 wt% Gly, orange line). In (**b**, **c**, **d**): deconvoluted FTIR spectra in the amide I region 1580–1700 cm⁻¹. The black dotted lines are experimental base-line corrected FTIR data.

applications[15,37]. This is particularly important when protecting oxygen-sensitive products from oxidation, which include food products like meats, dairy, and oils that are prone to spoilage in the presence of oxygen[38]. In Fig. 5, the oxygen permeability coefficient (OP) of five commercial plastic films is compared with that of the MB-based film containing 25 wt% glycerol. The average permeability coefficient for the MB film was measured at $0.33 \pm 0.7$ cm³ mm m⁻² day⁻¹ atm⁻¹(mean ± SD), demonstrating a significantly lower permeability than that of the commercial polymers PS, PE, PP and PET at 50% relative humidity (RH) and 23 °C. The reported oxygen permeability coefficient values (OP) for the MB-based films is an average of five measurements (0, 0, 0.095, 0 and 1.5 cm³ mm m⁻² day⁻¹ atm⁻¹). We note that three of the OP measurements (0 cm³ mm m⁻² day⁻¹ atm⁻¹) are below the oxygen permeance detection limit of the instrument. If one instead assumes an oxygen permeance of 0.2 cm³ m⁻² day⁻¹ atm⁻¹ for these three films, the average OP value will be $0.35 \pm 0.66$ cm³ mm m⁻² day⁻¹ atm⁻¹.

We hypothesize that the low OP of the MB-based films is due to the limited plasticizer content (25 wt% glycerol), which remained below the percolation threshold for glycerol[39]. This hypothesis was further supported by increasing the glycerol content to 30 wt%, which resulted in a significantly higher average OP of $16.9 \pm 4.2$ cm³ mm m⁻² day⁻¹ atm⁻¹ under identical experimental conditions (50% RH, 23 °C). Interestingly the OP value at 30 wt% glycerol loading is comparable to the oxygen permeability coefficient range of 17.7–20.8 cm³ mm m⁻² day⁻¹ atm⁻¹ (50% RH, 23 °C), previously reported for plasticized single-cell protein-based films containing 30 wt% glycerol[16]. The reported oxygen permeability coefficient values for the MB-based film at 30 wt% glycerol is an average of three measurements (15.7, 21.5 and 13.4).

The average OP ($0.33 \pm 0.7$ cm³ mm m⁻² day⁻¹ atm⁻¹) for the MB film with 25 wt% glycerol fell within the range of OP values reported for

EVOH (0.04–0.4 cm³ mm m⁻² day⁻¹ atm⁻¹ at 50% RH and 23 °C)[4]. EVOH is widely used in food packaging because it extends the shelf life of oxygen-sensitive foods by minimizing oxygen permeance and thereby slowing down food degradation. The excellent oxygen barrier properties of EVOH are due to the presence of hydroxyl (-OH) groups in its molecular structure and impermeable crystals, the former performing effectively in blocking gas transport at low relative humidities[12]. Other relevant high oxygen barrier packaging polymers include poly(-vinylidene chloride) (PVDC) and poly(ethylene naphthalate) (PEN) that demonstrate OP values of 0.01–0.3 cm³ mm m⁻² day⁻¹ atm⁻¹ and 0.5 cm³ mm m⁻² day⁻¹ atm⁻¹, respectively[1]. Our MB film has an OP value that is lower than PEN and at the upper boundary value for PVDC. To conclude, the bacterial biomass films studied here show promise as efficient oxygen barrier alternatives to EVOH, PEN and PVDC.

## Conclusions
The ability to limit oxygen transfer, combined with their sustainability, positions protein films as a promising solution for reducing food waste and plastic pollution. In this study, protein-rich (68%, dry basis) microbial biomass was processed into film and tray prototypes via compression molding. Glycerol functioned effectively as a plasticizing agent for the microbial biomass (MB), enhancing its suitability for application in biofilms and tray formats. The application of spray-drying as a pre-processing step, effectively disrupted cellular morphology and improved material homogeneity, thereby offering a scalable and industrially relevant approach. Additionally, the films displayed intrinsic tackiness, which could be exploited in the design of resealable packaging systems. Mechanical characterization revealed elastoplastic behavior with an elongation at break

**Table 2 | Secondary structures of the proteins (Cho et al.[34]) present in the MB films and tray with different glycerol contents**

| | Secondary structure | Powder | MB Film | | MB Tray |
|---|---|---|---|---|---|
| | Glycerol content (wt%) | 25 | 25 | 20 | 20 |
| Position (cm⁻¹) | Assignment | Relative area of gaussian components (%) | | | |
| 1680 | β-Sheets, weakly hydrogen-bonded peptide groups | 6.5 (1.2) | n.a. | n.a. | 7.7 (1.9) |
| 1667 | β-Turns | 21.5 (2.6) | 26.8 (0.5) | 30.9 (0.4) | 23.0 (4.1) |
| 1658 | α-Helices | n.a. | n.a. | n.a. | n.a. |
| 1651 | α-Helices and random coils | 25.8 (2.3) | 12.1 (1.9) | 11.0 (1.1) | 25.6 (3.6) |
| 1644 | Unordered | n.a. | n.a. | n.a. | n.a. |
| 1634 | β-Sheets, weakly hydrogen-bonded peptide groups | 20.9 (3.0) | 11.8 (8.6) | 15.6 (5.6) | 22.0 (3.4) |
| 1625 | β- Sheets, strongly hydrogen-bonded peptide groups | n.a. | 2.2 (1.4) | 3.2 (1.2) | n.a. |
| 1618 | β- Sheets, strongly hydrogen-bonded peptide groups | 25.4 (4.2) | 47.1 (17.6) | 39.3 (10.7) | 21.6 (3.9) |
| | **Secondary structures summarized** | (%) | (%) | (%) | (%) |
| | β-Turns | 21.5 | 26.8 | 30.9 | 23.0 |
| | α-Helices and random coils/unordered | 25.8 | 12.1 | 11.0 | 25.6 |
| | β-Sheets, weakly hydrogen-bonded peptide | 27.4 | 11.8 | 15.6 | 29.7 |
| | β- Sheets, strongly hydrogen-bonded peptide groups | 25.4 | 49.3 | 42.5 | 21.6 |

n.a. not attained.
The values within parenthesis are the uncertainties of the optimized values.

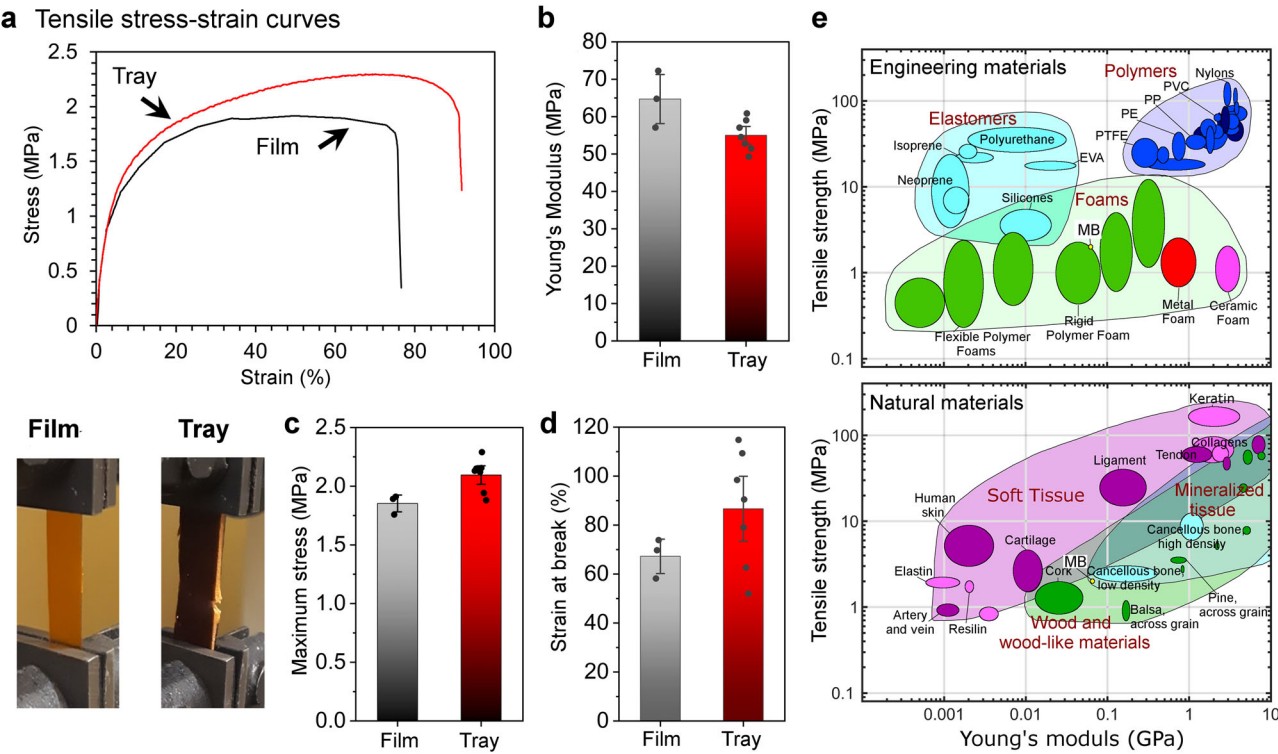

**Fig. 4 | Mechanical properties for microbial biomass (MB) based materials and other engineered or natural materials. a** Tensile stress-strain curves for the MB-based film (black line) and tray (red line), plasticized with 25 wt% and 20 wt% glycerol respectively. **b** Young's modulus, (**c**) maximum tensile stress and (**d**) strain-at-break. Average and standard error calculated from *n* = 3 (film) and *n* = 7 (tray). **e** Two Ashby plots, showing stiffness versus tensile strength properties for different engineering materials and natural materials, including the MB-based materials (yellow dots). Drawn after D. U. Shah[2].

(67–87%), maximal strength (≈2 MPa) and stiffness (55–65 MPa). These mechanical properties surpass those of previously reported single cell protein (SCP)-based films and are similar to those of films based on plasticized whey protein isolates. The plasticized MB demonstrated excellent oxygen barrier properties (0.33 ± 0.7 cm³ mm m⁻² day⁻¹ atm⁻¹) at standard testing conditions (50% RH and 23 °C). These values are comparable to those of ethylene vinyl alcohol (EVOH) layers (0.04–0.4 cm³ mm m⁻² day⁻¹ at

50% RH and 23 °C), which are widely utilized in multilayer packaging systems requiring high oxygen barrier properties.

Overall, the results demonstrate the feasibility of microbial biomass-derived films as sustainable and high-performance alternatives to petroleum-based polymers, particularly in packaging applications where high oxygen barrier properties are required. This work contributes to the advancement of circular bio-based materials and supports broader efforts

**Fig. 5 | Oxygen permeability coefficients for the microbial biomass (MB) films and commercial fossil-based materials. a** A microbial biomass-based film with 25 wt% glycerol. Scale bar: 1 cm. **b** Comparing the oxygen permeability coefficients (OP) of five commercial fossil-based materials (PS, PE, PP, PET and EVOH) and the MB-film. The OP ranges for PET, EVOH and the MB film are reported within parentheses. The average OP for the microbial biomass film was 0.33 cm$^3$ mm m$^{-2}$ day$^{-1}$ atm$^{-1}$ ($n = 5$). OP values for PS, PE, PP and PET are derived from Lange et al.[1]. OP for EVOH from Michiels et al.[4]. All oxygen permeability coefficients were obtained at 50% RH and 23 °C.

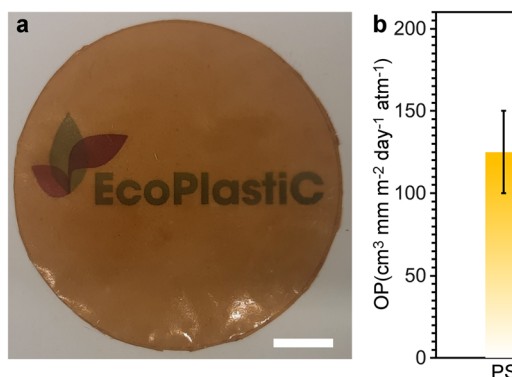
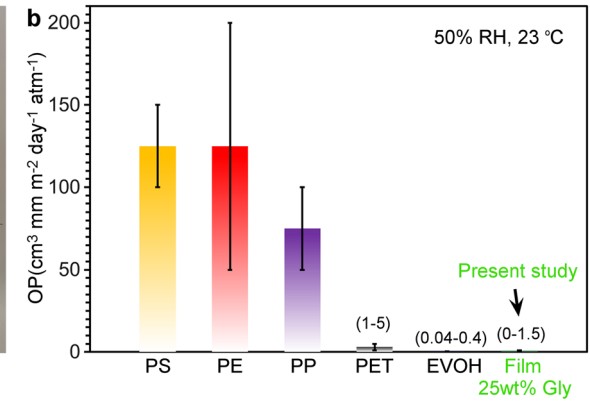

**Table 3 | Compression conditions for the films and tray**

| Biopolymer | Composition (%) | | Compression parameters | | | |
|---|---|---|---|---|---|---|
| Material | SCP (wt%) | Glycerol (wt%) | Press Temperature (°C) | Press Time (min) | Force (kN) | Thickness (mm) |
| Film[a] | 75 | 25 | 110 | 5 | 70-250 | 0.1-0.3 |
| Prototype Tray | 80 | 20 | 110 | 25 | 30 | ~2 |

[a]The same compression parameters were used for the MB films with 20 and 30 wt% glycerol.

aimed at reducing plastic waste and decreasing dependency on petrochemical-derived packaging systems.

## Materials and methods
### Materials
Spray dried protein rich microbial biomass (ValProMic) was obtained from Avecom Nv, Ghent, Belgium. The microbial biomass is derived from bacteria and is based on a patented process (BE 1021400)[40]. The bacteria were cultivated using liquid side streams from potato processing (process water from cutting potatoes into pieces). The composition of the biomass is presented in Supplementary Table 1 (Supplementary information). The biomass was stored in a vacuum desiccator until further processing. Glycerol (Gly) with 99% purity was purchased from Special Ingredients Europe (the Netherlands).

### Preparation of MB/glycerol blends
Glycerol was added in different proportions to the microbial biomass powder in a glass beaker and thoroughly mixed with a flat spatula until the mixture became lump-free. After mixing, blends were left to dry in a vacuum desiccator until further use.

### Production of MB films and tray
Preparation of MB films: The dried blend was evenly distributed into a rectangular mold (Steel, ca. 11 cm×7 cm, thickness 100 ± 7 μm,) and the sample mold was covered with a PET film on each side. The mold setup was then inserted between 3 mm metal plates, which were placed in a compression molder (Fontijne Press-TP 400, Netherlands) for the pressing operation. The pressing parameters are reported in Table 3. After hot-pressing (5 min), the machine was cooled down (from 110 °C) to room temperature utilizing an automatic water-cooling system with an average cooling rate of 24 °C min$^{-1}$. The obtained films had a thickness of ca. 0.1-0.3 mm. The films were stored in a vacuum desiccator prior to further testing.

Preparation of prototype tray: The blended powder was filled into the rectangular female steel mold, which was covered with a PET sheet to be able to remove the tray. The rectangular mold dimensions were 10.5 cm × 7.5 cm (Supplementary Fig. 2). A male mold covered also with a PET sheet was placed on top of the female mold. The whole assembly was placed into the

press that was preheated to 110 °C. Distance pins on the female part ensured that the final tray thickness became ca. 2 mm. The processing parameters are reported in Table 3. After the 25 min, the same cooling procedure as with the films was used.

Sealing of MB films: Two MB-films were placed in a Fermat 400 impulse sealer (Joke Mechanix, Germany), and heat-sealed together using minimum values of heat pulse time (1.5 s) and press time (1.5 s).

### Characterization
**Scanning electron microscopy (SEM)** of the surface and cross-sectional morphology of the samples were attained with a FE-SEM (Hitachi S-4800, Japan). Prior imaging, the samples were dried in a vacuum desiccator. Samples were mounted on a conductive carbon adhesive tape and sputter-coated with Cr using an Agar high-resolution sputter coater (model 208H) for 30 s.

**UV−Vis spectrophotometry** was performed using a UV−Vis spectrophotometer (Shimadzu UV 2550, Japan). The total transmittance measurements were conducted utilizing the ISR-2200 integrating sphere and BaSO$_4$ as the standard white reference.

**Tensile measurements** were assessed with an Instron 5966 device (Instron), equipped with 100 kN or 500 kN load cells. Specimens in the form of strips (length 70 mm, width 10 mm) were prepared and tested at a crosshead speed of 5 mm min$^{-1}$ and with a grip separation of 20 mm. The sample thickness was determined by taking an average of three measurements. The results were reported as mean values and standard deviations (or standard errors) from at least three measurements.

**FTIR** Fourier-transform infrared (FTIR) spectroscopy analyses of the films and tray were performed using a Perkin Elmer Spectrum 100 FT-IR spectrometer in attenuated total reflection mode. Spectra were recorded from 600−4000 cm$^{-1}$ with 32 scans and a resolution of 4 cm$^{-1}$. Prior to each measurement, the films were placed in a vacuum desiccator for a minimum of 48 h. The amide I region (1700−1580 cm$^{-1}$) was deconvoluted using the Multiple Peak Fit function (iteration algorithm: Levenberg Marquardt) in Origin 2020. Each spectrum of the amide I region (1700−1580 cm$^{-1}$) was first baseline corrected. During optimization the positions of the centres of the Gaussian-shaped peaks were fixed.

**Oxygen permeability** The oxygen permeability coefficients were attained using a PermeO$_2$ (ExtraSolution, Italy) that complies with ASTM

D3985. Measurements were performed using 99.9995% specialty grade oxygen, at 50% RH, 23 °C and 1 bar (partial pressure difference). The MB-films with 25 wt% glycerol and 30 wt% glycerol had a thickness of $0.240 \pm 0.080$ mm and $0.220 \pm 0.050$ mm, respectively. These were masked with aluminum foil with a 1.6 cm diameter hole (area 2.01 cm$^2$). For masked films, the instrument has a lower detection limit for the oxygen permeance of 0.2 cm$^3$ m$^{-2}$ day$^{-1}$ atm$^{-1}$.

**Thermal Gravimetrical Analysis (TGA)** was performed with a TGA-Q5000 instrument, using a heating rate of 10 °C' min$^{-1}$ from 30 °C to 800 °C. A modified atmosphere (100% N$_2$) was used with a gas flow rate of 50 mL min$^{-1}$.

## Data availability
The authors declare that the relevant data supporting the findings of this study are available within the paper and its supplementary information files. Numerical source data for graphs (FTIR, Mechanical and OP data) are provided as Supplementary Data 1. All data are available from the corresponding authors upon reasonable request.

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

## Acknowledgements

The EcoPlastiC project is funded by the European Union's Horizon Europe EIC Pathfinder programme under grant agreement No 101046758. The microbial biomass (MB) image in Fig. 1a was created with BioRender.com.

## Author contributions

K.R.B., K.W. and H.K. prepared MB materials. K.R.B. and E.G. performed TGA, UV-Vis, SEM and mechanical tests and analysis. B.S.C., C.V. and M.B.F. performed and analyzed Oxygen Permeability tests. M.H. performed the tackiness and sealing experiments. A.J.S. performed FT-IR and UV-vis measurements and analysis, wrote the first draft and made all illustrations. M.H. and A.J.S. supervised the project and K.R.B. contributed to manuscript writing. All authors have read and revised the manuscript.

## Funding

## Competing interests

The authors declare no competing interests.

## Additional information

**Peer review information** *Communications Chemistry* thanks the anonymous reviewers for their contribution to the peer reiew of this work. Peer reviewer reports are available.

