## [Transparent Peer Review file · Communications Chemistry]

High Oxygen Barrier Packaging Materials from Protein-rich Single-Celled Organisms

Corresponding Author: Professor Anna Svagan

Version 0:

Reviewer comments:

Reviewer #1

(Remarks to the Author)

this paper focuses on bioplastic films and trays for packaging from protein-rich microbial biomass using glycerol as the plasticizer. The biomass showed excellent film-forming through compression molding, resulting in materials with strong mechanical and gas barrier properties.

Here are comments to consider for improving the paper

1. How is this work different from the previously published work of the same group?

<https://pubs.acs.org/doi/full/10.1021/acssuschemeng.4c05739>

The methodology is the same, but only the application (oxygen barrier) differs.

2. EVOH and PET are good standards for oxygen barrier polymers, but a comparison to PVDC (polyvinylidene chloride) and PEN (polyethylene naphthalate) should be provided.

3. Glycerol is a widely used plasticizer in various biomaterials. However, its hygroscopic nature can alter material properties, resulting in increased flexibility, as FTIR data, which shows significant water content. Additionally, glycerol may negatively impact the oxygen barrier properties of materials. The authors should consider this issue. It might also be beneficial to explore other plasticizers for data comparison, as a comprehensive list of plasticizers suitable for proteins can be found in Krochta et al., Journal of Food Science, vol. 73, 2, 2008.

4. Figure 4 illustrates the mechanical properties of the bio-film, revealing modulus values that are ten times lower and strength values that are 50 to 100 times lower than those of synthetic polymers. Additionally, I find the sentence "comparable to plastics based on biomass DNA" unclear.

Reviewer #2

(Remarks to the Author)

Introduction needs to highlight the novelty of this study:

1. some data is outdated, for example " In 2010 alone, an estimated 4 to 12 million tons of plastic waste generated on land entered the marine environment."

2.the source of the protein-rich microbial (yeast, bacteria, microalgae, fungi) biomass should be clarified.What does protein-rich mean? what's the ratio of protein in the biomass? what's the role of the microbes in the biomass and how they affect the packaging material made from the biomass?

Results and Discussion

1. what type of microbes was used for cultivation? Is this process the same as fermentation of the liquid side streams from potato processing? What's the content of the liquid side streams from potato processing? Will different microbe yield different fermentation product?

2.The tensile strength of the plastic film is very low, why? Did you try lowering the plasticizer loading?

3.is the film humidity sensitive? How the properties change with moisture?

4. Is the film vulnerable to microbial contamination?

Conclusion

should include overlook of the work

Version 1:

Reviewer comments:

Reviewer #1

(Remarks to the Author)
the revision is satisfactory

We would like to express our appreciation to the two reviewers for their helpful comments on our manuscript. We realize that all of you have spent a considerable amount of time carefully reading the manuscript and formulating a strategy of action for us to make this paper better. We are very grateful for this.

We have addressed all the issues raised in the review and revised the manuscript along the lines suggested, and our revisions are in blue. Our responses to all the reviewers' comments are provided below.

The authors

Reviewer #1

this paper focuses on bioplastic films and trays for packaging from protein-rich microbial biomass using glycerol as the plasticizer. The biomass showed excellent film-forming through compression molding, resulting in materials with strong mechanical and gas barrier properties.

Here are comments to consider for improving the paper

1. How is this work different from the previously published work of the same group? <https://pubs.acs.org/doi/full/10.1021/acssuschemeng.4c05739>_The methodology is the same, but only the application (oxygen barrier) differs.

Answer: Both works explore the use of single-cell protein (SCP) or microbial biomass for bioplastic production. However, they differ significantly in their objectives, materials, processing methods, and focus areas. The present work focuses on engineering high-performance packaging films from protein-rich microbial biomass, achieving significantly enhanced mechanical properties (an order of magnitude higher strength and modulus, strain at break better than for materials in A. Bjurström et al.) and a low oxygen permeability ($\approx 0.33 \text{ cm}^3\text{-mm/m}^2\text{-day-atm}$), suitable for replacing ethylene vinyl alcohol (EVOH) in multilayer food packaging applications. In contrast, the study by A. Bjurström et al. prioritizes environmental sustainability, utilizing single-cell protein (SCP) sourced from cheese whey and anaerobic digestate to produce fully biodegradable films with high biogas recovery potential under anaerobic conditions. The present work employs spray-drying as a scalable preprocessing technique and demonstrates enhanced structural integrity, while the Bjurström study relies on cryogenic grinding, resulting in films with lower cohesion due to intact cellular structures and increased porosity. These findings illustrate a trade-off between functional performance and end-of-life environmental benefits, highlighting two complementary pathways for advancing sustainable biopolymer technologies. We have added a couple of new sentences in the introduction, page 3-4:

“Bjurström et al. developed microbial biomass-based films and evaluated their mechanical properties, moisture uptake behaviour, and biodegradability in the context of circular bioeconomy applications. However, the films exhibited poor mechanical performance due to low internal cohesion, and the authors highlighted the importance of employing the correct *pre-processing* step during preparation.

In this study, we explore the potential of developing mechanically improved packaging materials....”

2. EVOH and PET are good standards for oxygen barrier polymers, but a comparison to PVDC (polyvinylidene chloride) and PEN (polyethylene naphthalate) should be provided.

Answer: Thank you for this valuable suggestion! Following the reviewer's suggestion, a comparison to PVDC and PEN has been included in the results and discussion part in the manuscript. PEN has an OP of $\approx 0.5 \text{ cm}^3 \text{ mm m}^{-2} \text{ day}^{-1} \text{ atm}^{-1}$ whereas PVDC has an OP of 0.01-0.3 $\text{cm}^3 \text{ mm m}^{-2} \text{ day}^{-1} \text{ atm}^{-1}$ at 50% RH and 23 °C (Lange and Wyser Packaging Tech Science, Vol 15, pp 149-158). OP range for PVDC is comparable to that of EVOH. The OP of our protein-based material is lower than PEN and at the upper boundary value for PVDC. We have added this information to the manuscript, page 15:

"Other relevant high oxygen barrier packaging polymers include poly(vinylidene chloride) (PVDC) and poly(ethylene naphthalate) (PEN) that demonstrate OP values of 0.01 – 0.3 $\text{cm}^3 \text{ mm m}^{-2} \text{ day}^{-1} \text{ atm}^{-1}$ and 0.5 $\text{cm}^3 \text{ mm m}^{-2} \text{ day}^{-1} \text{ atm}^{-1}$, respectively.¹ Our MB film has an OP value that is lower than PEN and at the upper boundary value for PVDC. To conclude, the bacterial biomass films studied here show promise as efficient oxygen barrier alternatives to EVOH, PEN and PVDC."

3. Glycerol is a widely used plasticizer in various biomaterials. However, its hygroscopic nature can alter material properties, resulting in increased flexibility, as FTIR data, which shows significant water content. Additionally, glycerol may negatively impact the oxygen barrier properties of materials. The authors should consider this issue. It might also be beneficial to explore other plasticizers for data comparison, as a comprehensive list of plasticizers suitable for proteins can be found in Krochta et al., Journal of Food Science, vol. 73, 2, 2008.

Answer: We agree with the reviewer that the type of plasticizer introduced can significantly influence the material properties. However, our previous studies have shown that glycerol is the best plasticizer for protein-based materials (see Plasticizers for Protein-based materials, Ullsten et al. <http://dx.doi.org/10.5772/64073>). Therefore, glycerol was selected here. We also optimized the glycerol content (15, 20, 25 and 30 wt% tested) and ensured that the glycerol-containing MB materials were conditioned at 50% relative humidity prior to mechanical testing and oxygen barrier measurements. As a result, both glycerol and water (another plasticizer) were present in the materials during the assessment of mechanical and oxygen barrier properties. We have added a comment in the manuscript to highlight the importance of selecting the correct plasticizer, page 7:

"Although various plasticizers can be used with proteins,³⁰ we found, based on a large set of potential and known plasticizers, that glycerol was the most effective one for wheat gluten protein. Hence, it was used in the present study.³¹"

We also note that 25wt% glycerol was too high when processing the tray, but, on the other hand 20 wt% glycerol worked better, see our remark in the manuscript (page 7):

"Initially, the tray was fabricated using a 25 wt% glycerol-blended MB material. However, that tray lost its original shape within a month, indicating the need for increased stiffness. By reducing the glycerol content to 20 wt%, a tray with sufficient stiffness over time, was successfully produced."

4. Figure 4 illustrates the mechanical properties of the bio-film, revealing modulus values that are ten times lower and strength values that are 50 to 100 times lower than those of synthetic polymers. Additionally, I find the sentence “comparable to plastics based on biomass DNA” unclear.

Answer: Elastic moduli, tensile strength, and elongation at break for these films are comparable to those of plasticized protein isolate films, e.g. whey protein isolate films ($E \approx 50$ MPa, maximum tensile strength ≈ 4 MPa and elongation at break $\approx 100\%$, found in Krochta et al., Journal of Food Science, vol. 73, 2, 2008). Hence, the present MB films are not expected to outperform synthetic polymer-based materials. This implies that in applications where additional strength is needed, the MB films need to be sandwiched in a multilayer structure, similar to EVOH, using biodegradable plastics in the sandwich formation (lamellar structure). We have added a sentence on page 13:

“The elastic moduli (65 ± 8 MPa and 55 ± 4 MPa), maximum tensile stress (1.9 ± 0.1 MPa and 2.1 ± 0.1 MPa), and elongation at break for the MB-based film and tray are comparable to those of plasticized protein isolate films, e.g. whey protein isolate films ($E \approx 50$ MPa, maximum tensile strength ≈ 4 MPa and elongation at break $\approx 100\%$).³⁰”

“This suggests that for applications requiring enhanced mechanical strength, MB films should be incorporated into a multilayer structure, analogous to the configuration typically employed with EVOH.”

The sentence “comparable to plastics based on biomass DNA” has been removed, and instead has been replaced with the more relevant comparison to plasticized protein isolate films, see above.

Reviewer #2

Introduction needs to highlight the novelty of this study:

1. some data is outdated, for example " In 2010 alone, an estimated 4 to 12 million tons of plastic waste generated on land entered the marine environment."

Answer: Following the reviewer’s suggestion, the numbers and reference has been updated to more recent numbers (report published in 2021 by the UN Environment program), page 2:

“According to the UN Environment program, an estimated 11 million tonnes of plastic generated on land annually enter the world’s oceans.¹¹”

2.the source of the protein-rich microbial (yeast, bacteria, microalgae, fungi) biomass should be clarified. What does protein-rich mean? what's the ratio of protein in the biomass? what's the role of the microbes in the biomass and how they affect the packaging material made from the biomass?

Answer: The microbial biomass is derived from bacteria (see Claim 4 of the patent). It is based on a patented process (BE1021400 Verbeterde werkwijze voor winning van proteïnen uit proceswater). We have added this information to the experimental section (page 16).

The term "protein-rich" refers to the microbial biomass being enriched in protein, comprising approximately 60–70% of the dry matter - in this case, 68.4% protein on a dry matter basis, as reported in Table 1 in the manuscript. We have added this number in the abstract and where it is first mentioned in the introduction as well.

The protein-enriched microbial biomass was utilized in its dried form; thus, the microbial cells are inactive and do not play any active role in the resulting packaging material. After spray-drying it is no longer possible to visually identify any bacteria in the powder, see Figure 1c (I, MB Powder).

Results and Discussion

1. what type of microbes was used for cultivation? Is this process the same as fermentation of the liquid side streams from potato processing? What's the content of the liquid side streams from potato processing? Will different microbe yield different fermentation product?

Answer: The production is based on a patented process (BE1021400 "Verbeterde werkwijze voor winning van proteïnen uit proceswater"). The microbiome that was used, is an enriched microbiome that is originating from an enrichment culture.

The process water of the potato processing company mainly contains diluted starch that is currently not recovered from this side stream by the potato processing company and treated in the wastewater treatment plant. This is mainly the process water from cutting the potatoes into pieces.

Yes, it is expected that different microbes will yield differences in the product, this is expected as the composition (protein, lipids, carbohydrates) should vary between communities. However, in this process an optimal enriched microbiome was obtained in order to produce a microbial biomass with a high protein content.

2. The tensile strength of the plastic film is very low, why? Did you try lowering the plasticizer loading?

Answer: Elastic moduli, tensile strength, and elongation at break for these films are comparable to those of plasticized protein isolate films, e.g. whey protein isolate films ($E \approx 50$ MPa, maximum tensile strength ≈ 4 MPa and elongation at break $\approx 100\%$, found in Krochta et al., Journal of Food Science, vol. 73, 2, 2008). Also, the current MB films perform better than previous films made from yeast cells or single-cell proteins. We have further commented on these aspects in the manuscript, page 13:

"The elastic moduli (65 ± 8 MPa and 55 ± 4 MPa), maximum tensile stress (1.9 ± 0.1 MPa and 2.1 ± 0.1 MPa), and elongation at break for the MB-based film and tray are comparable to those of plasticized protein isolate films, e.g. whey protein isolate films ($E \approx 50$ MPa, maximum tensile strength ≈ 4 MPa and elongation at break $\approx 100\%$).^{30"}

"This suggests that for applications requiring enhanced mechanical strength, MB films should be incorporated into a multilayer structure, analogous to the configuration typically employed with EVOH."

Yes, we have tried a lower plasticizer loading of 15-20%. However, the films became brittle, which was undesired here.

3. is the film humidity sensitive? How the properties change with moisture?

Answer: In the present study we take moisture into account by following testing standards, and therefore we condition the films for 24h at 50 % RH prior to tensile testing. Additionally, oxygen barrier properties are measured for films equilibrated at 50% RH and 23 °C. Indeed, water acts as a plasticizer and influences film properties (mechanical and barrier) and the plasticizing effect will be a function of water content.

To limit and reduce any change in properties with higher moisture contents, these MB films should be incorporated into a multilayer structure, analogous to the configuration typically employed with EVOH – note that EVOH is also sensitive to moisture.

4. Is the film vulnerable to microbial contamination?

Answer: We expect the films to be degraded by microbes, hence the films must be protected from microbial contamination during their shelf-life, e.g. by limiting direct food-contact (lamellar structure as for EVOH).

Conclusion

should include overlook of the work

Answer: Following the reviewer's suggestion we have now included additional information in the conclusions section, page 15-16:

“The ability to limit oxygen transfer, combined with their sustainability, positions protein films as a promising solution for reducing food waste and plastic pollution. In this study, protein-rich (68%, dry basis) microbial biomass was processed into film and tray prototypes via compression molding. Glycerol functioned effectively as a plasticizing agent for the microbial biomass (MB), enhancing its suitability for application in biofilms and tray formats. The application of spray-drying as a pre-processing step, effectively disrupted cellular morphology and improved material homogeneity, thereby offering a scalable and industrially relevant approach. Additionally, the films displayed intrinsic tackiness, which could be exploited in the design of resealable packaging systems. Mechanical characterization revealed elastoplastic behavior with an elongation at break (67–87%), maximal strength (≈ 2 MPa) and stiffness (55-65 MPa). These mechanical properties surpass those of previously reported single cell protein (SCP)-based films and are similar to those of films based on plasticized whey protein isolates. The plasticized MB demonstrated excellent oxygen barrier properties ($0.33 \pm 0.7 \text{ cm}^3 \text{ mm m}^{-2} \text{ day}^{-1} \text{ atm}^{-1}$) at standard testing conditions (50% RH and 23 °C). These values are comparable to those of ethylene vinyl alcohol (EVOH) layers ($0.04 - 0.4 \text{ cm}^3 \cdot \text{mm} \cdot \text{m}^{-2} \cdot \text{day}^{-1} \cdot \text{atm}^{-1}$ at 50% RH and 23 °C), which are widely utilized in multilayer packaging systems requiring high oxygen barrier properties.

Overall, the results demonstrate the feasibility of microbial biomass-derived films as sustainable and high-performance alternatives to petroleum-based polymers, particularly in packaging applications where high oxygen barrier properties are required. This work contributes to the advancement of circular bio-based materials and supports broader efforts aimed at reducing plastic waste and decreasing dependency on petrochemical-derived packaging systems.”

Dear Editor,

We have addressed all the Editorial requests and revised the manuscript along the lines suggested, and our revisions in the manuscript are in red.

Sincerely,

Anna J. Svagan (on behalf of all the authors)

EDITORIAL REQUESTS

* Your manuscript should comply with our policies and format requirements, detailed in our style and formatting guide (<https://www.nature.com/documents/commsj-phys-style-formatting-guide-accept.pdf>).

Answer: The manuscript has been edited and complies with the policies and format requirements.

* Please edit your manuscript according to the editorial requests in the attached table, and outline revisions made in the right hand column. If you have any questions or concerns about any of our requests, please do not hesitate to contact me. It is important that each request be addressed in order to avoid delays in accepting your manuscript. Please upload the completed table with your manuscript files as a Related Manuscript file.

Answer: The manuscript has been edited according to the editorial requests. The completed table is uploaded as a related manuscript file.

* Nature journals require authors of life sciences research papers to include relevant details about several elements of experimental and analytical design in their manuscripts. This initiative aims to improve the transparency of reporting and the reproducibility of published results and is described at: www.nature.com/authors/policies/reporting.pdf. To ensure that your manuscript complies with our policy, please pay close attention to the 'methods' and 'legends' sections of our checklist for authors: Reporting requirements for life sciences research. You may also find the following collection of articles on statistics for biologists helpful: Statistics for Biologists.

Answer: The Method section in the manuscript contains relevant details to reproduce the published results. However, we note that our paper should not be considered as a "true" life science research paper because we use a commercially sourced single-cell protein product in the preparation of our material. We provide the manufacturer's specifications of the raw material in the manuscript.

REVIEWERS' COMMENTS:

Reviewer #1 (Remarks to the Author):

the revision is satisfactory

Answer: We thank reviewer #1 for the positive evaluation of our work.